# Evaluation of Changes in the Cardiac Function before and after Transcatheter Edge-to-Edge Mitral Valve Repair in Healthy Dogs: Conventional and Novel Echocardiography

**DOI:** 10.3390/ani12010056

**Published:** 2021-12-28

**Authors:** Kenta Sasaki, Danfu Ma, Ahmed S. Mandour, Yusuke Ozai, Tomohiko Yoshida, Katsuhiro Matsuura, Aki Takeuchi, Chieh-Jen Cheng, Hussein M. El-Husseiny, Hanan Hendawy, Kazumi Shimada, Lina Hamabe, Akiko Uemura, Ryou Tanaka

**Affiliations:** 1Department of Veterinary Surgery, Faculty of Veterinary Medicine, Tokyo University of Agriculture and Technology, Fuchu-shi 183-8509, Tokyo, Japan; vm.knt.sasaki@gmail.com (K.S.); dandanma1000@gmail.com (D.M.); ozaivet@icloud.com (Y.O.); tomohiko7731-yoshida@yahoo.co.jp (T.Y.); k.matsuura.vet@gmail.com (K.M.); akkiki89@gmail.com (A.T.); john_199328@yahoo.com.tw (C.-J.C.); hussien.alhussieny@fvtm.bu.edu.eg (H.M.E.-H.); hanan_attia@vet.suez.edu.eg (H.H.); ruiyue1221@gmail.com (K.S.); linahamabe@vet.ne.jp (L.H.); 2College of Veterinary Medicine, Nanjing Agricultural University, No. 1 Wei-Gang, Xuanwu District, Nanjing 210095, China; 3Department of Animal Medicine (Internal Medicine), Faculty of Veterinary Medicine, Suez Canal University, Ismailia 41522, Egypt; 4Department of Surgery, Anesthesiology, and Radiology, Faculty of Veterinary Medicine, Benha University, Moshtohor, Toukh, Elqaliobiya 13736, Egypt; 5Department of Veterinary Surgery, Anesthesiology and Radiology, Faculty of Veterinary Medicine, Suez Canal University, Ismailia 41522, Egypt; 6Department of Veterinary Surgery, Division of Veterinary Research, Obihiro University of Agriculture and Veterinary Medicine, Obihiro 080-8555, Hokkaido, Japan; anco@vet.ne.jp

**Keywords:** TEER, mitral valve repair, transcatheter mitral valve intervention, echocardiography

## Abstract

**Simple Summary:**

Mitral valve regurgitation in dogs is a common cardiac disease. The standard treatment for this disease is medical therapy, but as the condition progresses, it leads to congestive heart failure. In recent years, surgical treatment of mitral valve regurgitation in dogs has emerged, but the recommended procedure is invasive and time-consuming. Therefore, a less invasive and time-consuming surgical procedure is necessary for future utilization in the treatment of mitral valve regurgitation. Recently, a novel transcatheter edge-to-edge mitral valve repair device became available and is anticipated to be a minimally invasive and less time-consuming device, but there are no data available on the hemodynamic changes after its cardiac implantation.

**Abstract:**

Mitral valve regurgitation is a common canine heart disease. Transcatheter Edge-to-Edge Repair (TEER) is a transcatheter, edge-to-edge mitral repair device that uses a hybrid approach. No detailed information has been published on the hemodynamic effect of TEER on cardiac function. The aim of this report is to provide a longitudinal observation of the cardiac functional changes observed after TEER implantation in normal dogs using traditional, two-dimensional speckle tracking, and color M-mode echocardiographic methods. In the current report, TEER was implanted into two healthy dogs under general anesthesia. An echocardiographic examination was performed at baseline and weekly postoperative follow-ups were conducted until the fourth week. Successful TEER implantation was achieved with a short operation time (98 and 63 min) in the two dogs. Functional mitral valve regurgitation, elevated E/e’ ratio, elevated radial strain, and stable intraventricular pressure gradients (IVPG) were observed after the operation in the dogs. Mild non progressive mitral valve stenosis was observed in both dogs. TEER is a minimally invasive method for mitral valve surgery that necessitates more clinical trials. With longitudinal observation of heart function using novel approaches, better outcomes will be expected.

## 1. Introduction

Mitral valve disease (MVD) is the most common cardiac disease in dogs, especially in small breeds. Up to 75% of dogs with clinical signs of congestive heart failure (CHF) suffer from mitral valve regurgitation (MR) which is caused by the myxomatous degeneration of the valve leaflets and chordae tendineae [1]. The recommended treatment for MR has changed over the years. Medical treatment remains the first-line treatment option and aim to alleviate the clinical signs and postpone the onset of CHF. However, the long-term prognosis is poor since medical treatment cannot reverse the pathological changes to the mitral valve apparatus [1,2,3,4]. Several in vivo and clinical trial studies have explored the surgical correction of the damaged mitral valve (MV). In the early 2000s, MV replacement surgeries were performed [5,6,7]. However, the high incidence of prosthetic valve thrombosis limited their long-term outcomes. Recently, MV repair under cardiopulmonary bypass has been suggested as a promising treatment option, and several commercial facilities in Japan now offer this treatment [4,8,9,10]. The outcome associated with MV repair appears to be excellent; however, the costs and highly invasive nature of the surgery are viewed as problematic for patient owners.

The surgical procedures to manage this condition in humans require cardiopulmonary bypass and have a high risk of complications. In human medicine, percutaneous MV repair using the MitraClip® system (Abbott Vascular, Abbott Park, IL, USA) was established in 2008 and has been demonstrated to be safe, ameliorates clinical symptoms and reduces the durations of hospitalization and mortality by addressing severe MV functional defects in high-risk surgical patients suffering from CHF [11]. Unfortunately, MitraClip is not available for use in canine patients, especially in small dogs, due to the device size. To apply a treatment effect similar to that provided by MitraClip in small-sized dogs, a percutaneous approach should be modified to fit the cardiac and vessel sizes of these small-sized dogs. A hybrid approach is a procedure that combines a conventional surgical procedure with an interventional procedure, such as the use of a catheter-based procedure guided by fluoroscopy or transesophageal echocardiography (TEE) imaging. The hybrid approach has previously been used in a puppy to treat congenital heart disease. This technique does not require extracorporeal circulation and is less invasive than traditional open-heart surgery [12,13].

Transcatheter Edge-to-Edge Repair (TEER) is an easy-to-operate, transcatheter MV repair system for use in humans and dogs using a hybrid approach [14]. This device appears to be effective; however, no detailed information has been published except for a simple prognostic evaluation in a single canine and porcine study [14,15]. To evaluate the prognostic effect of the treatment on the heart, echocardiographic changes should be evaluated.

Recent echocardiographic approaches such as two-dimensional speckle tracking echocardiography (2D-STE) and intraventricular pressure gradients (IVPG) calculated from color M-mode echocardiography create a breakthrough in cardiology research for the early diagnosis of cardiac dysfunctions, as compared with other traditional methods [16,17,18]. In the present report, the echocardiographic changes were evaluated using conventional echocardiography, 2D-STE and CMME, before and weekly after the implantation of the TEER in two clinically normal dogs.

## 2. Materials and Methods

### 2.1. Surgical Procedure

Implantation of the TEER was performed in two female beagles, aged 2 and 2.5 years and weighing 8.6 and 10.3 kg, respectively. These two dogs were healthy, based on physical examinations, hematobiochemical profile, conventional echocardiography, and urinalysis. The detailed procedure of the surgical approach has been previously described [15]. Briefly, the dogs were maintained on general anesthesia using 2–3% isoflurane (Pfizer, Tokyo, Japan) after induction with propofol (Fresenius Kabi, Tokyo, Japan). The TEER was guided by a transesophageal echocardiography (TEE) probe in the median longitudinal and transverse views [15]. Heparinized normal saline (100 IU/kg) was administered intravenously after exposing the heart at the left sixth intercostal space via thoracotomy. The pericardium was incised and the apical area leaked. After placing a purse-string on the cardiac apex, the TEER MV repair delivery system was inserted through the cardiac apex first into the left ventricle and then into the left atrium under echocardiographic guidance. The MV was captured, and the TEER was deployed after the clamp position was confirmed (Figure 1). The delivery system was released, and the cardiac apex were sutured by the purse-string. Pericardium, thoracic wall and skin were sutured by 3-0 Prolene, PDS and Nylon, respectively. No medication was given to the dogs during the operation. Postoperative recoveries were monitored by a critical care team consisting of licensed veterinarians. During the postoperative period, clinical signs, echocardiography, electrocardiogram, blood pressure, blood picture, serum chemistry, and urine volume were monitored [9]. Additional proper administration will be executed by the critical care team, but in these two dogs, we did not prescribe any drugs. Postoperative antibiotics (ampicillin, Meiji Seika, Tokyo, Japan) and painkillers (buprenorphine hydrochloride, Otsuka Pharmaceutical, Tokyo, Japan) were administered for 3 days. Clopidogrel (Nichi-Iko, Tokyo, Japan) was administered at 2 mg/kg for 1 week postoperatively.

### 2.2. Echocardiography

#### 2.2.1. Conventional Echocardiography

Echocardiographic examination was performed using a Hitachi ProSound Premier 75CV (Hitachi Aloka Medical, Tokyo, Japan) or Hitachi LISENDO 880 (Hitachi Aloka Medical, Japan) ultrasound system equipped with a 1–15 MHz TEE probe. Standard conventional echocardiography from both right and left sides including the ordinary long axis and short-axis views was performed for each dog at the baseline (preoperative) and 3 times postoperative (at 1-, 2-, and 4- weeks) according to De Madron et al. [19]. Echocardiography was performed twice every week and the average was obtained. From the right parasternal short-axis view, M-mode evaluation of the left ventricular function at the papillary muscle level was obtained from 5 selected images. Measurements include LV end-diastolic and end-systolic diameters (LVIDd, LVIDs), interventricular septal thickness in diastole and systole (IVSd, IVSs), LV free wall thickness in diastole and systole (LVPWd, LVPWs) and fraction shortening (FS%). At the level of the heart base, by two dimensional echocardiography, maximal left atrium-to-aortic diameter ratio (LA/Ao) was evaluated at the closing of the aortic valve [20]. On the left side, assessment of trans-mitral flow by pulsed-wave Doppler was performed to measure early (E) and late (A) inflow waves. Continuous Doppler was used to evaluate mitral valve regurgitation (MR velocity, MR dP/dt). The mitral orifice anterioposterior commissural axis (MOOA) and the mitral orifice commissural axis (MOCA) from the left parasternal apical two and four chamber view were obtained [21]. Tissue Doppler imaging (TDI) at the point of attachment of the mitral valve with septal and lateral walls of the LV was reported and the TDI parameters, including systolic (s) and diastolic indices (e’, a’, e’/a’) at both sides of the base of the mitral valve were evaluated.

#### 2.2.2. Two-Dimensional Speckle Tracking Echocardiography (2D-STE)

The 2D-STE was performed from the right parasternal short-axis view at the papillary muscle level and left parasternal apical four-chamber view. Digital cine-loops of three consecutive heart cycles with the frame rate set at 70–110 frames/s were captured from all dogs and saved as DICOM files for further processing using in-house code software (DAS-RS1 software 1.1v, Hitachi Aloka Medical). By using DAS-RS1, the short axis analysis and free use methods of video processing were used for short axis and long axis 2DSTE evaluation, respectively. The endocardium and the epicardium were outlined automatically at end-systole and end-diastole. The tracing borders were manually edited to ensure that the visualized real wall motion and the whole wall thickness were incorporated into the region of interest used in the frame-by-frame basis analysis. From the short-axis view, segmentation with an angular interval of 60° divided the LV into the following 6 segments; anterior, anterior septal, anterior lateral, inferior, inferior septal, inferior lateral. Based on tracing of these 6 segments, we reported the left ventricular global circumferential strain (GCS), global circumferential strain rate (GCSR) and global radial strain (GRS). From the left apical four-chamber view, longitudinal strain (LS) and longitudinal strain rate (LSR) were assessed [22].

#### 2.2.3. Color M-Mode Echocardiography (CMME)

Intraventricular pressure gradient (IVPG) was obtained from the apical four-chamber view through CMME and analyzed with homemade software in MATLAB [17]. Details of the used procedure and method of calculation of IVPG have previously been described [11]. Total IVPG, basal IVPG, and mid-to-apical IVPG were reported.

## 3. Results

The operation times for each operation were 98 and 63 min. The dogs recovered from the general anesthesia uneventfully, and no adverse cardiac symptoms from the operation were observed. Three weeks after TEER implantation, dog 1 was diagnosed with mild pulmonary edema by chest X-ray, however, because no related symptoms were observed, medication was not prescribed. No arrhythmia was observed during the operation. 

There was a single ventricular premature complex (VPC) after surgery in both cases; however, no treatment was required because the related symptoms were not observed and there were no cardiac disturbances observed after VPC.

### 3.1. Echocardiographic Measurement

#### 3.1.1. Conventional Echocardiography

The detailed echocardiographic parameters are listed in Table 1. The left ventricular dimensions showed no dramatic changes after the surgeries. The fractional shortening was increased postoperatively in both dogs compared with the baseline, but this increase was much higher in dog 2. The mitral inflow velocity waves (E and A) were significantly increased after the operation and throughout the investigation period (Figure 2). The mitral valve orifice in both dogs was slightly narrowed after the operations (2–3 mm difference) at the second and third weeks. In addition, the stenosis was not progressive and both dogs showed slight enhancement of the mitral orifice at the fourth week. Concomitantly, the LA/Ao, early diastolic inflow to myocardial velocity (E/e’), and mitral valve regurgitation velocity (MR) were increased in both dogs after surgery.

#### 3.1.2. Two-Dimensional Speckle Tracking and Color M-Mode Echocardiography

Table 2 summarizes the 2D-STE (Figure 3) and CMME (Figure 4) data in the two operated dogs. The radial strain was increased in both dogs during the observation period. The total and basal IVPGs were only increased in the first dog at week 1 postoperative, then they reduced; meanwhile; in the second dog, the total and basal IVPGs remained higher than the baseline until week 4 post-operative. The mid-to-apical IVPGs were found to be elevated in the dog 1 until week 4 compared with the baseline; however, it was reduced in dog 2.

## 4. Discussion

MVD is the most common cardiac disease in dogs. The current approach for the surgical treatment of MVD is valvuloplasty under extracorporeal circulation conditions, which is greatly invasive and time-consuming [9,10]. Instead, the hybrid approach represents a promising technique in veterinary medicine because it is less invasive and more convenient than the other available surgical options.

In the current report, the TEER, an example of a minimally invasive device, was successfully implanted in two healthy dogs. The operation time was approximately one hour to one and a half hours, which is much shorter than the time required for valvuloplasty under extracorporeal circulation conditions [15,23,24]. Additionally, the postoperative recovery was rapid, and less postoperative management was required. The surgical time needed in the second case was a half an hour less than in the first case, hence, the team became accustomed to the technical procedures of the device. Collectively, in the future, the above mentioned information indicates that the TEER could allow multiple operations to be performed in one day.

The use of TEE is necessary to guide device implantation. In this report, device implantation was performed using three-dimensional (3D) TEE in one case and with 2D TEE in the other case. The use of 3D TEE is advantageous for the detection of the position of the device in the heart; however, 2D TEE can also be used, although changing the view is necessary for successful implantation. Taking into consideration that 3D TEE is expensive, the use of 2D TEE is more realistic and it is more likely to be adopted by more facilities.

Similarly to the MitraClip, the TEER clamps the anterior and posterior mitral leaflets to reduce the regurgitation [15]. The clamping arms of the TEER close toward the central line, resulting in tighter clamping compared to the MitraClip. Therefore, the effect of the TEER includes mitral annuloplasty, which represents a fundamental component of MV repair. Although mitral annuloplasty can reduce MR, it can also cause mitral stenosis [25]. In the current study, mild stenosis of the MV was observed in both cases at one week post-operative, and was gradually reinstated later on. However, the reported values of the mitral annular orifice were within the normal range in normal dogs [26]. In a previous study performed in a swine model, an echocardiographic evaluation was conducted postoperatively; however, no detailed Doppler study was conducted [14]. Compared with the swine model, the effect of the implantation of the device is expected to be more significant in the canine model. The influence of TEER implantation on the heart should be evaluated in greater detail in the canine model.

E- and A-waves have been reported to serve as indicators of increased left ventricle filling pressures and diastolic dysfunction, respectively [1]. The prediction of postoperative echocardiographic changes in normal dogs will offer important information regarding the evaluation of postoperative hemodynamic changes. In the present report, the postoperative E wave was higher than the preoperative measurement, likely caused by the decreased mitral orifice area following the device placement. The E wave is sometimes used for the estimation of atrial pressure; however, the E wave can be affected by multiple factors, including preload, diastolic function, and others [18,27]. 

The E/e’ ratio has been reported to correlate with the pulmonary capillary wedge pressure when the mitral inflow passes through the MV [28]. With the implantation of TEER, the mitral inflow will pass through MV from two independent orifices, and the decreased size of the mitral orifice will benefit MR affected dogs by suppressing volume overload. However, this outcome is associated with a risk of mitral stenosis, and the mitral inflow velocity (peak E) may become elevated due to the mitral orifice narrowing. In the two dogs from this report, narrowing of the orifice resulted in functional mitral valve regurgitation. The functional MR velocity indicates the pressure difference between LA and LV in the systole. The elevation of MR velocity after the procedure illustrates that the LV pressure in the systole was elevated, which is correlated with the elevation of FS and GRS [29,30].

In the two dogs, we observed an increase in FS% and radial strain. FS is widely used to evaluate the systolic function in dogs. However, FS does not accurately detect it because it only observes myocardial contraction in a radial direction at a pair of specific myocardial segments, whereas in fact, myocardial shortening occurs in different directions [31]. In addition, myocardial deformation is important because it evaluate the myocardial contraction from base to apex (longitudinal strain), systolic shortening in the short-axis view (circumferential strain), and myocardial thickening from the endocardium to epicardium (radial strain). This allows interpretation of regional and global abnormalities with high predictive ability [16,32]. As the dogs included in this study are healthy and have the normal reference values of these parameters before surgery, the increased radial strain and FS% may be due to the induced cardiac wound, which stimulates the heart to work more to survive. An increase in cardiac contractility does not always suggest improvement in heart function, as radial and circumferential deformation could be increased to compensate for LV longitudinal dysfunction [33,34].

The correlation between E/e’ and the pulmonary capillary wedge pressure can be challenging in patients with altered MV geometry. IVPG is the force that suctions the blood from the left atrium into the left ventricle during early diastole. The basal part of IVPG has been shown to correlate with left atrium pressure, and mid-to-apical IVPG has been correlated with myocardium movement [35]. In dog 1, the hemodynamic parameters changed dramatically, with peak E-wave value at 2 weeks postoperation nearly four-fold higher than that measured at the baseline, which indicated elevated left atrium pressure caused by mitral stenosis. The pulmonary edema observed at 2 weeks postoperation in dog 1 was also confirmed to be due to hemodynamic changes. In dog 2, pulmonary edema was not observed, despite a high E/e’ ratio. For the evaluation of congestion after TEER implantation, basal IVPG appears to be a more reliable measure than E/e’.

The current report has some limitations. This report only includes two healthy dogs. The occurrence of clinical abnormalities, including mitral stenosis, is expected because the orifice was normal before surgery. In addition, this approach might be helpful in mitral valve disease, which is associated with the large MV orifice observed in MMVD. Currently, surgical correction is the only option for fixing the enlarged MV orifice in clinic. TEER provided us with a fixing tool that is easy to manipulate. The previous study proved the efficiency of the device in the case of MMVD [15]. However, in our study, we only used healthy dogs and report the conventional and novel echocardiography, which shows the novelty of this report. Although the data could not be conclusive, they provide a delightful in-sight into the new approach for mitral valve repair in dogs.

## 5. Conclusions

TEER provides a minimally invasive method for mitral valve repair that warrants further clinical trials in dogs with mitral valve disease to suppress volume overload. The narrowing of the mitral valve, caused by TEER, might be beneficial to MMVD patients. Basal IVPG appears to be a potential index for evaluating postoperative congestion.

## Figures and Tables

**Figure 1 animals-12-00056-f001:**
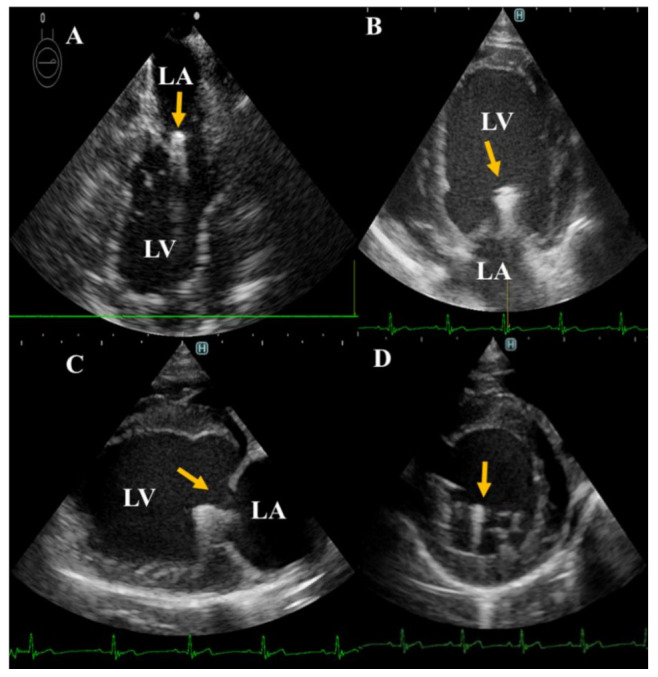
Two-dimensional echocardiographic localization of the implanted TEER (yellow arrowhead). Successful localization of TEER during operation through transesophageal-guided echocardiography (**A**). For post-operative follow-up, transthoracic echocardiography from left apical (**B**), right long axis (**C**), and right short-axis views (**D**) were used. LA left atrium; LV, left ventricle.

**Figure 2 animals-12-00056-f002:**
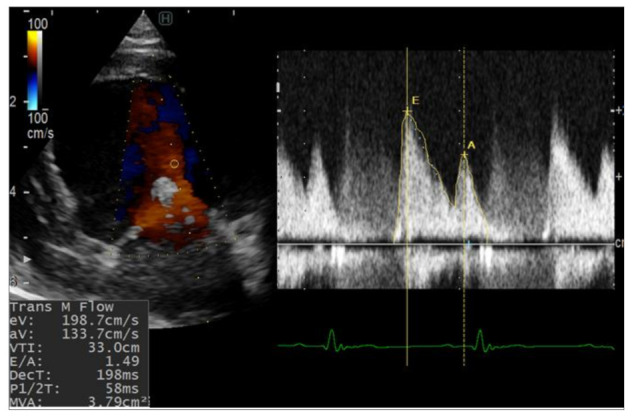
Pulsed-wave Doppler echocardiography for evaluation of the mitral inflow from left apical four-chamber view after TEER implantation. Increase in E wave values was observed after TEER implantation. eV, early diastolic filling velocity; aV, atrial filling velocity.

**Figure 3 animals-12-00056-f003:**
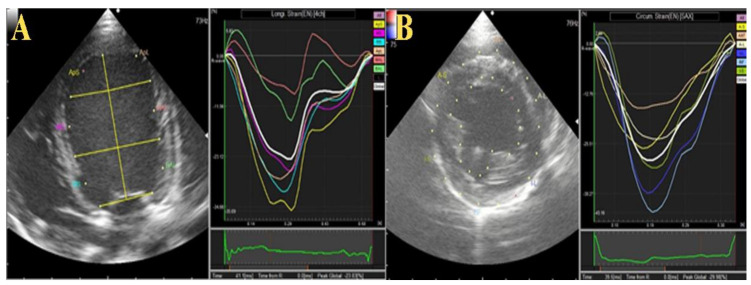
Schematic illustration of two-dimensional speckle tracking echocardiography (2DSTE) in the operated dogs. The entire left ventricular endocardium and epicardium were traced for measurement. (**A**) longitudinal strain was obtained from the left apical four chamber view. (**B**) circumferential strain obtained from short-axis view.

**Figure 4 animals-12-00056-f004:**
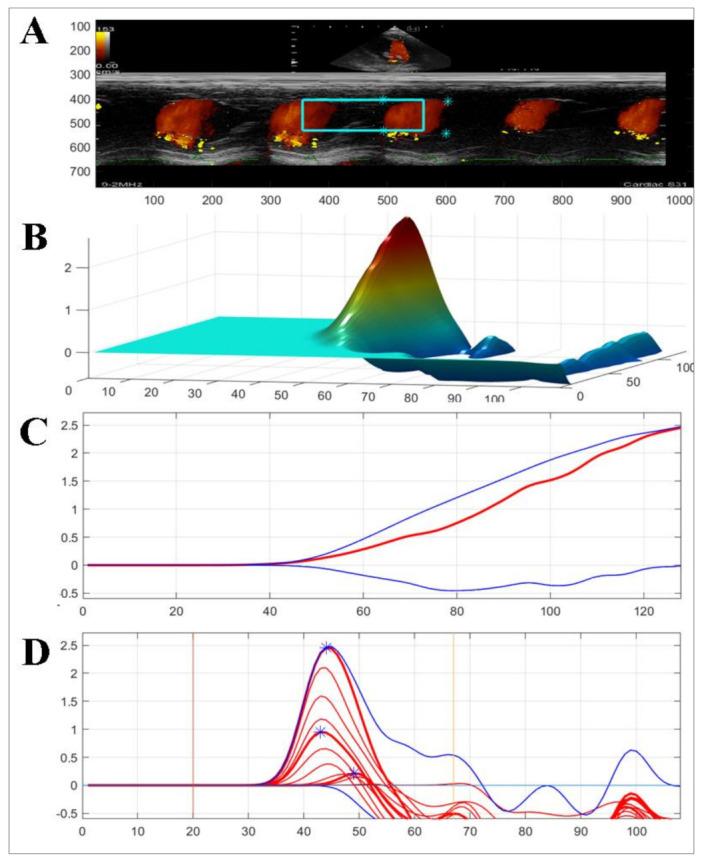
Representation of color M-mode echocardiography used to measure the intraventricular pressure gradients (IVPG) from left apical four-chamber view. Firstly, the mitral inflow was observed, then color M-mode echocardiography (CMME) was switched on and photos were captured for further calculation MATLAB. (**A**) color M-mode echocardiography; (**B**) three-dimensional temporal and spatial profiles of the left ventricular IVPG; (**C**,**D**) IVPG in early diastole-, the top (blue), middle (red), and bottom (blue) lines represent inertial, total, and convective IVPG, respectively.

**Table 1 animals-12-00056-t001:** Sequential echocardiographic and basic evaluation before and after TEER.

Parameters	1st Dog				2nd Dog			
	Pre	1 w	2 w	4 w	Pre	1 w	2 w	4 w
HR (bpm)	142	162	168	108	116	136	157	112
IVSd (mm)	7.7	8.9	6.6	7.4	6.6	10.2	7.2	7.2
LVIDd (mm)	34.5	29.8	33.9	34.7	26.9	35.4	33.4	35.0
LVPWd (mm)	6.2	7.2	7.0	6.4	13.9	7.9	8.1	7.0
IVSs (mm)	10.8	12.2	12.9	11.5	11.3	11.5	11.9	12.2
LVIDs (mm)	19.8	15.7	16.1	17.6	20.5	23.5	14.5	16.8
LVPWs (mm)	10.0	11.3	12.5	11.2	11.9	11.8	14.9	13.4
FS %	42.6	47.3	52.6	49.2	23.8	34.1	56.6	52.0
E (cm/s)	64.8	151.5	215.4	198.7	59.9	98.6	127.2	113.9
A (cm/s)	22.1	102.1	101.2	133.7	36.8	100.3	85.9	87.9
S’ (cm/s)	7.9	9.4	8.0	7.9	11.5	7.5	9.5	7.6
é (cm/s)	10.6	8.9	7.4	6.3	10.0	8.6	8.8	9.5
á (cm/s)	7.5	8.1	6.6	5.8	9.5	8.4	7.3	7.9
E/é (cm/s)	6.1	17.0	29.1	31.5	6.0	11.5	14.5	12.1
MR velocity (cm/s)	-	484.8	461.2	564.8	-	479.8	561.9	576.5
MR dP/dt max	-	1340	846	3217	-	5358	4021	4021
LA/Ao	1.0	1.2	1.1	1.3	1.1	1.1	1.1	1.1
MOAA (mm)	24.2	22.1	22.5	23.3	23.8	20.6	20.8	22.3
MOCA (mm)	19.9	19.0	23.0	23.5	20.7	20.6	20.4	22.5
SAP	105	115	112	135	110	112	132	134
DAP	85	90	85	91	86	81	93	89
MAP	98	107	103	120	102	102	119	119

Echocardiographic assessment of cardiac function in the operated dogs before and after TEER through short-axis view (M-mode), mitral inflow, and tissue Doppler imaging. HR, heart rate; IVSd, interventricular septum diastolic diameter; LVPWd, LV free wall diastolic diameter; LVIDd, left ventricular (LV) internal diastolic diameter; IVSs, interventricular septum systolic diameter; LVPWs, LV free wall systolic diameter; LVISd, LV internal systolic diameter; FS, fraction shortening; LA/Ao, left atrial to aortic diameter ratio; E, early diastolic velocity; A, atrial contraction flow velocity; S’, systolic annular velocity of the LV wall; e’, early diastolic tissue velocity; a’, late diastolic annular velocity of the LV wall; E/e’, early diastolic mitral velocity to the early diastolic velocity of the LV wall ratio; MR, mitral regurgitation; MOACA, mitral orifice anterioposterior commissural axis; MOCA, mitral orifice commissural axis; SAP, systolic arterial pressure; DAP, diastolic arterial pressure; MAP, mean arterial pressure.

**Table 2 animals-12-00056-t002:** Two-dimensional speckle tracking and color M-mode echocardiography evaluation of the operated dogs before and after TEER.

Variables	1st Dog				2nd Dog			
	Pre	1 w	2 w	4 w	Pre	1 w	2 w	4 w
Speckle tracking echocardiography								
GRS %	16.8	18.5	13.2	19.7	10.3	12.0	12.6	13.2
GCS %	−16.0	−13.4	−14.8	−14.4	−14.8	−14.4	−17.3	−19.7
GCSR %	−2.2	−1.5	−1.8	−2.2	−2.2	−2.1	−2.2	−2.5
LS %	−17.5	−14.4	−15.5	−15.7	−23.8	−21.3	−23.7	−26.9
LSR %	1.4	1.4	2.4	2.8	3.0	2.2	2.2	1.9
Color M-mode echocardiography								
Total IVPG (mmHg)	0.75	1.26	0.66	0.73	0.91	0.93	0.98	0.95
Basal IVPG (mmHg)	0.44	0.6	0.23	0.23	0.43	0.63	0.66	0.59
Mid-apical IVPG (mmHg)	0.31	0.66	0.43	0.5	0.48	0.30	0.31	0.35

GRS, global radial strain; GCS, global circumferential strain; GCSR, global circumferential strain rate; LS, longitudinal strain; LSR, longitudinal strain rate.

## Data Availability

The data presented in this study are available on request.

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
