# Peer review of "Evaluation of Changes in the Cardiac Function before and after Transcatheter Edge-to-Edge Mitral Valve Repair in Healthy Dogs: Conventional and Novel Echocardiography"

_animals, 2021, doi:10.3390/ani12010056_

Round 1

Reviewer 1 Report

This is an interesting manuscript describing the application of a hybrid procedure to clamp the mitral valve leaflets in healthy dogs as a possible treatment for MMVD in dogs. However, the methods are only partially described, and the discussions and conclusions do not fully comment the reported results.

INTRODUCTION: 

LINE 51: "valve leaflets or chordae tendineae" should be replaced with "valve leaflets and chordae tendineae"

LINE 55: "valve leaflets or chordae tendineae" can be replaced with "mitral valve apparatus"

MATERIALS AND METHODS:

The authors should explain in detail the execution of the procedure: did they incise the pericardium? I assume that the device was inserted through the cardiac apex first into the left ventricle and then into the left atrium under echocardiographic guidance. Was the pericardium also sutured after the procedure?

FIGURE 1, LINES 112-114: Authors wrote: “During operation, concomitant transesophageal echocardiography and transthoracic echocardiography from left apical view, right long axis, and right short-axis views were used”: Authors should explain how did they perform the left apical transthoracic projection during surgery.

LINES 120-128: Authors should provide adequate reference of what do they mean with “full conventional echocardiography” in the dog. Which method do they use to evaluate the LA/Ao ratio? They should provide adequate references.

Authors should describe in detail or provide adequate references on the execution of the M-mode and right ventricular outflow tract velocity measurements from the right short axis view. They should also provide adequate references on the mitral inflow, aortic flow, and tissue Doppler imaging (TDI) assessment from the left apical view.

The echocardiographic variables were measured once or more times in order to report the mean values? Did they perform echocardiography with simultaneous ECG or was the dog connected to an ECG during the procedure? Which parameters of left ventricular function did they take into consideration? This should be specified also in M&M not only in results.

LINE 132: which left apical view? Two, three, four or five chambers view?

RESULTS

Were the dogs monitored with ECG or Holter after the procedure?

Did dogs have arrhythmias during the procedure or after? I suppose yes. Which arrhythmias? How did you control it?

Did authors monitor the echocardiographic evolution of the ventricular lesion?

LINES 153-154: How did they make the diagnosis of pulmonary edema? How did they treat it?

LINES 162-163: how did authors measure the MV orifice? they should describe how they did this measurement in the M&M section.

TABLE 1: “MR volocity” should be replaced by “MR velocity”. Authors should describe in the M&M section how did they obtain the MR velocity, the MR dp/dt, the MOOA and the MOCA

TABLE 2: the abbreviation for Global Longitudinal Strain used in the M&M section (GLS) does not correspond to that used in table 2. The abbreviations of global circumferential strain, global circumferential strain rate, global radial strain and global radial strain rate should be GCS, GCSR, GRS and GRSR, respectively.

DISCUSSION

LINES 234-236: Authors should provide adequate reference about the duration of the valvuloplasty under extracorporeal circulation.

LINES 236-237: Authors should specify what did they monitor during the post operative period. In M&M and in RESULS section there is no description of this. They should provide adequate reference about the duration of postoperative recovery after valvuloplasty under extracorporeal circulation.

LINE 243: in the M&M section authors should describe the use of 3D TEE and the measurements they assessed with it, with adequate references. The results of 3D TEE measurements should be reported in the RESULTS section.

LINES 249-257. based on authors results Valve Clamp creates severe mitral stenosis and severe increase of left atrial pressure in both patients. As a result, in 1 of the two dogs pulmonary edema was detected 3 weeks after surgery. As one of the most important complications of the degenerative mitral valve disease is the increase of left atrial pressure, I think that this procedure is not very advantageous compared to other therapeutical options. Authors should discuss more about the hemodynamic and clinical negative effect of mitral valve stenosis caused by ValveClamp. Authors should compare their results with the results obtained in human medicine. Is mitral valve stenosis a frequent complication of this procedure? Which is the gold standard treatment for MR in humans? When is indicated to apply Valve Clamp in humans?

In 1 of the two dogs pulmonary edema was detected 3 weeks after surgery. Could the treatment have influenced the echocardiographic measurements performed at week 4?  Authors should discuss this aspect.

LINE 264: how did authors measure the mitral orifice area? the results of these measurements performed before and after the procedure should be reported in the RESULS section.

LINES 259-266: authors should provide adequate references about E and A in dogs.

Authors should discuss all the results they found:

Why did fractional shortening increase after the procedure?

Why did MR velocity increase after the procedure?

Why did radial strain increase after procedure?

In my opinion in the limitations of the study authors should state that the results are obtained only in 2 healthy dogs, and the results (severe mitral stenosis in both of them) do not support the possible application of this tool in myxomatous mitral valve disease. Other studies are needed to verify its utility in treatment of MMVD.

Author Response

Dear respected reviewer 1

We appreciate your effort in reviewing our manuscript. Here we addressed all reviewer’s comments and the reply point to point from the author’s point of view. Our answer is in red in the rebuttal and with the track change, red color and highlighted in the manuscript. We hope that our revision is sufficient for your question and suggestions.

Comments and replies to the reviewer 1

(x) English language and style are fine/minor spell check required

The entire manuscript has been revised for English by experienced coathors. We will do double English check in the next step if it is required.

INTRODUCTION:

  • LINE 51: "valve leaflets or chordae tendineae" should be replaced with "valve leaflets and chordae tendineae"  

I have corrected the manuscript. Please check Line 52.

  • LINE 55: "valve leaflets or chordae tendineae" can be replaced with "mitral valve apparatus"

I have corrected the manuscript. Please check Line 56.

MATERIALS AND METHODS:

  • The authors should explain in detail the execution of the procedure: did they incise the pericardium? I assume that the device was inserted through the cardiac apex first into the left ventricle and then into the left atrium under echocardiographic guidance. Was the pericardium also sutured after the procedure?

The description was added Line102

(The pericardium was incised and the apical area leaked. After placing a purse-string on the cardiac apex, the ValveClamp MV repair delivery system was inserted through the cardiac apex first into the left ventricle and then into the left atrium under echocardiographic guidance. The MV was captured, and the ValveClamp was deployed after the clamp position was confirmed (Figure 1). The delivery system was released, and the cardiac apex was sutured by the purse-string. Pericardium, thoracic wall, and skin were sutured by 3-0 Prolene, PDS and Nylon, respectively.).

  • FIGURE 1, LINES 112-114: Authors wrote: “During operation, concomitant transesophageal echocardiography and transthoracic echocardiography from left apical view, right long axis, and right short-axis views were used”: Authors should explain how did they perform the left apical transthoracic projection during surgery.

During operation, only transesophageal echocardiography was used. While preoperation and follow-up, the standard echocardiographic protocol was used. The word concomitant is a typing mistake. The text has been corrected as follows.

Two-dimensional echocardiographic localization of the implanted ValveClamp (yellow arrowhead). Successful localization of ValveClamp during operation through transesophageal guided echocardiography (A). For postoperative follow-up, transthoracic echocardiography from left apical (B), right long axis (C), and right short-axis views (D) were used. LA left atrium; LV, left ventricle.

  • LINES 120-128: Authors should provide adequate reference of what do they mean with “full conventional echocardiography” in the dog. Which method do they use to evaluate the LA/Ao ratio? They should provide adequate references.

We mean complete standard protocol. However, it has been deleted to avoid confusion.

The text was revised as follow (129-147):

Standard conventional echocardiography from both right and left sides including the ordinary long axis and short-axis views were performed for each dog at the baseline (pre-operative) and 3 times postoperative (at 1-, 2- and 4-weeks) according to De Madron et al. [21].

Maximum LA/Ao was reported at the closure of Ao valve. This reference was added (J Vet Cardiol. 2019 doi: 10.1016/j.jvc.2019.11.001)

  • Authors should describe in detail or provide adequate references on the execution of the M-mode and right ventricular outflow tract velocity measurements from the right short axis view. They should also provide adequate references on the mitral inflow, aortic flow, and tissue Doppler imaging (TDI) assessment from the left apical view.

More details were added and the references were also included. (READ PREVIOUS COMMENT) Line 129-147.

Echocardiography was performed twice every week and the average was obtained from 5 selected images. From the right parasternal short-axis view, M-mode evaluation of the left ventricular function at the papillary muscle level was obtained. Measurements include LV end-diastolic and end-systolic diameters (LVIDd, LVIDs), interventricular septal thickness in diastole and systole (IVSd, IVSs), LV free wall thickness in diastole and systole (LVPWd, LVPWs), and fraction shortening (FS%). At the level of the heart base, by two-dimensional echocardiography, the maximal left atrium to aortic diameter ratio (LA/Ao) was evaluated at the closing of the aortic valve [22]. On the left side, an assessment of trans-mitral flow by pulsed-wave Doppler was performed to measure early (E) and late (A) inflow waves. Continuous Doppler was used to evaluating mitral valve regurgitation (MR velocity, MR dp/dt). The Mitral orifice anterioposterior commissural axis (MOOA) and the Mitral orifice commissural axis (MOCA) from the left parasternal apical two and four-chamber view were obtained [23]. Tissue Doppler imaging (TDI) at the point of attachment of the mitral valve with septal and lateral walls of the LV were reported and the TDI parameters include systolic (s) and diastolic indices (e’, a’, e’/a’) at both sides of the base of the mitral valve were evaluated.

  • The echocardiographic variables were measured once or more times to report the mean values? Did they perform echocardiography with simultaneous ECG or was the dog connected to an ECG during the procedure? Which parameters of left ventricular function did they take into consideration? This should be specified also in M&M not only in results.

Echocardiography was performed twice every week and the average was obtained. From the right parasternal short-axis view, M-mode evaluation of the left ventricular function at the papillary muscle level was obtained from 5 selected images. (Line 132-135)

LINE 132: which left apical view? Two, three, four or five chambers view?

LINE 151 indicates the left apical four-chamber as in figure 3.

RESULTS

  • Were the dogs monitored with ECG or Holter after the procedure?

We monitored with ECG after procedure Line 106-107

  • Did dogs have arrhythmias during the procedure or after? I suppose yes. Which arrhythmias? How did you control it?

No arrhythmia was observed during the operation. There was a single VPC after surgery, but no treatment was required because it only showed once and the cardiac movement is in a satisfying status. (Line 177).

Did authors monitor the echocardiographic evaluation of the ventricular lesion?

We evaluated the heart function as a whole. We make strain analysis to specify each region's performance and the global strain and strain rate were taken (Table 2).

  • LINES 153-154: How did they make the diagnosis of pulmonary edema? How did they treat it?

Three weeks after ValveClamp implantation, dog 1 was diagnosed with mild pulmonary edema by chest X-ray. Treatment of pulmonary edema was not performed because the edema was mild and no related symptoms were observed.

The text was modified as follow: Three weeks after ValveClamp implantation, dog 1 was diagnosed with mild pulmonary edema by chest X-ray, because no related symptom was observed, medication was not prescribed (174-176).

  • LINES 162-163: how did authors measure the MV orifice? they should describe how they did this measurement in the M&M section. Authors should describe in the M&M section how did they obtain the MR velocity, the MR dp/dt, the MOOA and the MOCA

Continuous Doppler was used to evaluating mitral valve regurgitation (MR velocity, MR dp/dt). The Mitral orifice anterioposterior commissural axis (MOOA) and the Mitral orifice commissural axis (MOCA) from the left parasternal apical four and two-chamber view were obtained [1].

  • TABLE 1: “MR volocity” should be replaced by “MR velocity”.

It has been corrected

  • TABLE 2: the abbreviation for Global Longitudinal Strain used in the M&M section (GLS) does not correspond to that used in table 2. The abbreviations of global circumferential strain, global circumferential strain rate, global radial strain and global radial strain rate should be GCS, GCSR, GRS and GRSR, respectively.

The text has been corrected as the reviewer suggested (line 161-164; line 228-229).

A follow: Based on tracing of these 6 segments, we reported the left ventricular global circumferential strain (GCS), global circumferential strain rate (GCSR) and global radial strain (GRS). From the left apical four-chamber view, longitudinal strain (LS) and longitudinal strain rate (LSR) were assessed [24].

DISCUSSION

  • LINES 234-236: Authors should provide adequate reference about the duration of the valvuloplasty under extracorporeal circulation.

A reference was added (line 266)

  • LINES 236-237: Authors should specify what did they monitor during the post operative period. In M&M and in RESULTS section there is no description of this. They should provide adequate reference about the duration of postoperative recovery after valvuloplasty under extracorporeal circulation.

Postoperative recovery was monitored by a critical care team consisting of licensed veterinarians. During the postoperative period, clinical signs, echocardiography, electrocardiogram, blood pressure, blood picture, serum chemistry, and urine volume were monitored [10]. And proper administration will be executed by the critical care team, but in these two dogs, we did not prescribe any drugs.

This manuscript is short communication and the basic and well-known data did not add. 

  • LINE 243: in the M&M section authors should describe the use of 3D TEE and the measurements they assessed with it, with adequate references. The results of 3D TEE measurements should be reported in the RESULTS section. →

No data acquisition was collected during surgery by 3DTEE. At first, we were confused about the feasibility of 2D echo machine to guide us during the operation. So it was used as a guide and no measurements were taken because we investigate the measurements during follow-up.

  • LINES 249-257. based on authors results Valve Clamp creates severe mitral stenosis and severe increase of left atrial pressure in both patients. As a result, in 1 of the two dogs pulmonary edema was detected 3 weeks after surgery. As one of the most important complications of the degenerative mitral valve disease is the increase of left atrial pressure, I think that this procedure is not very advantageous compared to other therapeutical options.

Thank you for your comment. We considered it in the revised section. Here we have to consider that these dogs are healthy and showed no clinical abnormalities before experimental surgery. Therefore, the occurrence of clinical abnormalities, including mitral stenosis is expected because the orifice was normal before surgery. In addition, this approach might be helpful in mitral valve disease which is associated with large MV orifice in MMVD. And the elevation in this report is caused by the MR which comes from unsuitable MV orifice size, but in the MR dog, ValveClamp will make the MV orifice more suitable for the patients, in another word, ValveClamp will decrease the left atrial pressure in MR dogs. Currently, surgical correction is the only option in fixing the enlarged MV orifice in the clinic, ValveClamp provides us a fixing tool easy to manipulate.

This part has been considered in the revised manuscript (line 334-343).

 A previous study proved the efficiency of the device in case of MVD (https://pubmed.ncbi.nlm.nih.gov/33392290/). However, in our study, we used only healthy dogs and report the conventional and novel echocardiography in this study.

In addition, in our study, the mitral valve orifice in both dogs was slightly narrowed after the operations (2-3 mm difference) at the second and third weeks. In addition, the stenosis was not progressive and both dogs showed slight enhancement of the mitral orifice at the fourth week (178-190).

  • Authors should discuss more about the hemodynamic and clinical negative effect of mitral valve stenosis caused by ValveClamp. Authors should compare their results with the results obtained in human medicine. Is mitral valve stenosis a frequent complication of this procedure? Which is the gold standard treatment for MR in humans? When is indicated to apply Valve Clamp in humans?

The operation of ValveClamp in dogs is in the research stage, and this study was carried out for future development. Catheter therapy using mitral clip is used in humans, but cannot be used in dogs due to catheter size problems. Therefore, as an alternative to catheter therapy, this device treatment is considered. This study has been conducted only on two dogs. Another reviewer suggests not to discuss or compare our results with other data and we should present the data without directing the results to a certain way because it is only two cases. However, more details were added in the discussion (line 283-285; 309-322) and other parts from 323-334 were revised.

  • In 1 of the two dogs pulmonary edema was detected 3 weeks after surgery. Could the treatment have influenced the echocardiographic measurements performed at week 4?

Authors should discuss this aspect.

Pulmonary edema was mild and spontaneous recovery before the end of the fourth week. No specific treatment for edema was given. So we think it does not affect the measurements.

  • LINE 264: how did authors measure the mitral orifice area? The results of these measurements performed before and after the procedure should be reported in the RESULS section.

The Mitral orifice anterioposterior commissural axis (MOOA) and the Mitral orifice commissural axis (MOCA) from the left parasternal apical four and five-chamber view were obtained. Line 139.

The mitral valve orifice in both dogs was slightly narrowed after the operations (2-3 mm difference) at the second and third weeks. In addition, the stenosis was not progressive and both dogs showed slight enhancement of the mitral orifice at the fourth week. (line 187-190).

  • LINES 259-266: authors should provide adequate references about E and A in dogs.

Reference was added in Line 265

Schiller, N.B.; Shah, P.M.; Crawford, M.; DeMaria, A.; Devereux, R.; Feigenbaum, H.; Gutgesell, H.; Reichek, N.; Sahn, D.; Schnittger, I., et al. Recommendations for Quantitation of the Left Ventricle by Two-Dimensional Echocardiography. Journal of the American Society of Echocardiography 1989, 2, 358-367.

  • Authors should discuss all the results they found.

We appreciate the reviewer's concern. Another reviewer suggests not to discuss or compare our results with other data and we should present the data without directing the results to a certain way because it is only two cases. However, more details were added in the discussion (line 283-285; 309-322) and other parts from 323-334 were revised. 

  • Why did fractional shortening increase after the procedure? Why did radial strain increase after the procedure?

In the two dogs, we observed an increased FS% and radial strain. FA is widely used to evaluate the systolic function in dogs. However, FS does not accurately detect it because it only observes myocardial contraction in a radial direction at a pair of specific myocardial segments, where in fact, myocardial shortening occurs in different directions. In addition, myocardial deformation in three dimensions evaluating the myocardial contraction from base to apex (longitudinal strain), systolic shortening in short axis (circumferential strain), and myocardial thickening from endocardium to the epicardium (radial strain). This allows interpretation of regional and global abnormalities with high predictive ability. Concerning that dogs included in this study are healthy and have the normal reference values of these parameters before surgery, therefore, the increased radial strain and FS% may be due to the induced cardiac wound that stimulates the heart to work more to survive. The increase in cardiac contractility does not always suggest improvement of heart function as radial and circumferential deformation could be increased to compensate for LV longitudinal dysfunction.

  • Why did MR velocity increase after the procedure?

MR after the procedure was caused by the stretch of Valveclamp, the MR velocity indicates the pressure difference between LA and LV. The elevation of MR velocity only illustrates the LV pressure were elevated which is common in dogs (line 306-308).

  • In my opinion in the limitations of the study authors should state that the results are obtained only in 2 healthy dogs, and the results (severe mitral stenosis in both of them) do not support the possible application of this tool in myxomatous mitral valve disease. Other studies are needed to verify its utility in the treatment of MMVD.

Thank you for your comment. We considered it in the revised section. Here we have to consider that these dogs are healthy and showed no clinical abnormalities before experimental surgery. Therefore, the occurrence of clinical abnormalities, including mitral stenosis is expected because the orifice was normal before surgery. In addition, this approach might be helpful in mitral valve disease which is associated with large MV orifice in MMVD. And the elevation in this report is caused by the MR which comes from unsuitable MV orifice size, but ValveClamp will make the MV orifice more suitable for the patients in the MR dog. In another word, ValveClamp will decrease the left atrial pressure in MR dogs.  Currently, surgical correction is the only option in fixing the enlarged MV orifice in the clinic, ValveClamp provides us a fixing tool easy to manipulate.

This part has been considered in the revised manuscript (line 334-343).

The current report has some limitations. This report includes only two healthy dogs, the occurrence of clinical abnormalities, including mitral stenosis is expected because the orifice was normal before surgery. In addition, this approach might be helpful in mitral valve disease which is associated with large MV orifice in MMVD. Currently, surgical correction is the only option in fixing the enlarged MV orifice in the clinic, ValveClamp provides us a fixing tool easy to manipulate. A previous study proved the efficiency of the device in the case of MMVD[16]. However, in our study, we used only healthy dogs and report the conventional and novel echocardiography in this study, which is shown the novelty of this report. Although the data could not be conclusive, it provides a delighting step into the new approach for mitral valve repair in dogs.

Reviewer 2 Report

Dear Authors,

made revisions have improved your manuscript. I recommend the publication of the manuscript in Animals

Author Response

Dear respected reviewer

Thank you so much for your great effort. We appreciate your concern about our revision.

Reviewer 3 Report

I think that it is a very advanced article, the surgical technique, the quality of the equipment used and the strain measurements are incredible.

It should be noted in the title that the animals are healthy On the other hand, it seems strange to me that they use this technique
in healthy animals, in the end they have caused in both cases a
harmful iatrogenic mitral stenosis that has caused an
increase in pressure and an increase in mitral regurgitation,
which is what is being tried to remedy in this sickness.
Therefore, edema even forms in one of the animals.
It is adequate to describe the technique, but when doing it in healthy dogs and
obtaining these results they have not been able to demonstrate that it helps animals
suffering from EVMD
In Table 1, I would ask you to add the heart rate in each of the tests,
which helps to make sense of those values, since most of them will be influenced
by HR.

Author Response

Dear respected reviewer 3

We appreciate your effort in reviewing our manuscript. Here we addressed all reviewer’s comments and the reply point to point from the author’s point of view. We hope that our illustration is sufficient for your question and suggestions.

 Comments and reply to the reviewer 3

 It should be noted in the title that the animals are healthy On the other hand, it seems strange to me that they use this technique

The word healthy was added to the title.

in healthy animals, in the end they have caused in both cases a harmful iatrogenic mitral stenosis that has caused an increase in pressure and an increase in mitral regurgitation, 
which is what is being tried to remedy in this sickness. Therefore, edema even forms in one of the animals. 

It is adequate to describe the technique, but when doing it in healthy dogs and obtaining these results they have not been able to demonstrate that it helps animals suffering from EVMD.

We agree with the reviewer's concern. Here we have to consider that these dogs are healthy and showed no clinical abnormalities before experimental surgery. Therefore, the occurrence of clinical abnormalities, including mitral stenosis is expected because the orifice was normal before surgery. In addition, this approach might be helpful in mitral valve disease which is associated with a large MV orifice in which the surgical correction of the leakage requires to do narrowing of the orifice.

The current report has some limitations. This report includes only two healthy dogs, the occurrence of clinical abnormalities, including mitral stenosis is expected because the orifice was normal before surgery. In addition, this approach might be helpful in mitral valve disease which is associated with large MV orifice in MMVD. Currently, surgical correction is the only option in fixing the enlarged MV orifice in the clinic, ValveClamp provides us with a fixing tool easy to manipulate. A previous study proved the efficiency of the device in the case of MMVD[16]. However, in our study, we used only healthy dogs and report the conventional and novel echocardiography in this study, which is shown the novelty of this report. Although the data could not be conclusive, it provides a delighting step into the new approach for mitral valve repair in dogs.

In Table 1, I would ask you to add the heart rate in each of the tests,
which helps to make sense of those values, since most of them will be influenced 
by HR. Heart rate was added in table 1.

Round 2

Reviewer 1 Report

I thank the authors for making the required changes to the manuscript; however, there are still some minor revisions:
Line 309: FA should be replaced by FS
Lines 296-298: please provide references
Lines 299-300: please provide references
Line 306: "the pressure difference between LA and LV" should be replaced with "the pressure difference between LV and LA in systole"
Lines 307-308: I don't understand your explication for LV pressure increase. Please rephrase it clearly and provide references. I think that LV pressure increased because of FS and GRS increase.

Author Response

Dear reviewer,   Thank you for your detailed review. We revised the manuscript based on your advice, we rephrased some sentences, the citation has been added. And we also checked the grammar. Now the quality of this paper has been promoted.

Reviewer 3 Report

The manuscript has improved with the modifications made by the authors My recommendations have been incorporated and I agree

Author Response

Dear reviewer
Thank you for your detailed review. we appreciate your help to enhance the quality of the manuscript.

This manuscript is a resubmission of an earlier submission. The following is a list of the peer review reports and author responses from that submission.

Round 1

Reviewer 1 Report

I think it’s very important to study alternative methods of therapeutic treatment for mitral valve disease in dogs. So I consider this short communication very interesting.

There are several minor considerations:

Line 19: ValveClamp is in lower case letter whereas in the rest of the text this word is in capital letter.

Line 20: there is a comma after a full stop.

Line 113: Figure ValveClump instead of ValveClamp

Mayor considerations:

Line 65 to line 69: These descriptions do not correspond to surgical procedures.

Line 82: Figure 1. It is not clear what type of echo modality the authors have used: TTE or TEE?. I think the images correspond to TTE echo, if I am right, then the exact form to describe the view will be: right parasternal long axis view and right parasternal short axis view.

Line 84: It is not clear what echo modality the authors have used at baseline or other times. It is not clear if the dogs were awake or under general anaesthesia when the echo were performed.

Line 102: Results: LA/Ao ratio increased in dog 1 over time, maybe this data is no statistically significant but there is an increase. MR velocity increased in dog 2 over time and in dog 1 there’s an increase compared with the first measurement. So It is too conclusive to say: “no significant hemodynamic effect”, even more if one dog develop a pulmonary oedema.

Line 103: The E/e’ ratio increased in the two dog, not just in dog 2. And following with the previous comment, this results show hemodynamic effects in my opinion.

Line 132: In the surgical procedures do not figure these considerations: three-dimensional echo or TEE, please complete surgical procedures section.

Line 144: They inform of mitral stenosis caused by Valve Clamp. I would like to know the explanation, because there are no signs of mitral stenosis in similar studies even long time studies.

Line 155: That’s right, E wave is used for the estimation of atrial pressure, in this specific case it could be reflect the high left atrial pressure in a dog which develops a pulmonary oedema.

Line 161:  Just as it happen in these two dogs. Both showed severe increase in peak E.

Line 181: “….to mitral stenosis in normal dogs, whereas this outcome may benefit MR dogs……”. I don’t think this could be a good conclusion. One out of two normal dogs developed a pulmonary oedema probably due to mitral stenosis. In a MR dog, with left atrial volume overload, a mitral stenosis doesn’t seem to me a good condition.

On the other hand, similar studies with MR B1 dogs( which they are primarily normal dog from a hemodynamic point of view)  do not show mitral stenosis after ValveClamp implantation.

In general, I suggest to report the data without trying to give a false idea of a good solution for MR dogs. The data come from just 2 normal dogs, and one developed pulmonary oedema, so I think the report must be just a reflection of the title, it may explain the effect such as they occurred.

These do not mean ValveClamp couldn’t be useful, just reflect the data the author achieved.

I encourage the authors to change the approach to the article. It is good to have reports of new system to treat mitral valve disease in dogs.

Reviewer 2 Report

The study of new surgical procedures for the therapy of myxomatous mitral valve disease (MMVD) in dogs is of fundamental importance for the advancement of veterinary cardiology and we will hear more often in the future. However, despite the high scientific value of these studies, in this manuscript there are many weaknesses that led me to consider it as not publishable in this journal:
• The introduction does not adequately describe the background on this topic and does not contain adequate bibliographic references on MMVD;
• The materials and methods are not adequately described to allow the reproducibility of the study;
• The results are presented in insufficient detail;
• The discussion is poor in content and references to previous studies, there is no analysis of the study limits;

Reviewer 3 Report

Dear Authors,

The aim of the study was to describe a new surgical technique in the treatment of mitral valve regurgitation on dogs affected by myxomatous mitral valve disease.

The manuscript is well written and organized. No information about the signalment of dogs (breed), history, stage of cardiac heart failure and treatment performed are included. Moreover, the manuscript report only two cases of ValveClamp implantation, and in my opinion is very poor to perform the discussion about the possible application of this tool in myxomatous mitral valve disease. So, I encourage the Authors to extend the cases number to submit again the manuscript.